# Cardiac afferent signals can facilitate visual dominance in binocular rivalry

John P Veillette*, Fan Gao, Howard C Nusbaum

Department of Psychology, University of Chicago, Chicago, United States

**Abstract** Sensory signals from the body's visceral organs (e.g. the heart) can robustly influence the perception of exteroceptive sensations. This interoceptive–exteroceptive interaction has been argued to underlie self-awareness by situating one's perceptual awareness of exteroceptive stimuli in the context of one's internal state, but studies probing cardiac influences on visual awareness have yielded conflicting findings. In this study, we presented separate grating stimuli to each of subjects' eyes as in a classic binocular rivalry paradigm – measuring the duration for which each stimulus dominates in perception. However, we caused the gratings to 'pulse' at specific times relative to subjects' real-time electrocardiogram, manipulating whether pulses occurred during cardiac systole, when baroreceptors signal to the brain that the heart has contracted, or in diastole when baroreceptors are silent. The influential 'Baroreceptor Hypothesis' predicts the effect of baroreceptive input on visual perception should be uniformly suppressive. In contrast, we observed that dominance durations increased for systole-entrained stimuli, inconsistent with the Baroreceptor Hypothesis. Furthermore, we show that this cardiac-dependent rivalry effect is preserved in subjects who are at-chance discriminating between systole-entrained and diastole-presented stimuli in a separate interoceptive awareness task, suggesting that our results are not dependent on conscious access to heartbeat sensations.

**\*For correspondence:**
johnv@uchicago.edu

**Competing interest:** The authors declare that no competing interests exist.

## eLife assessment

This is a binocular rivalry study that uses ECG to present visual stimuli pulsing in line with cardiac events, to examine whether systole-entrained stimuli (i.e. presented during the period where the heart has contracted) are suppressed within visual awareness. Arguably out of line with this idea, the dominance durations were increased for systole-entrained stimuli. The manuscript addresses an **important**, precisely defined, and theoretically well-motivated question using sophisticated experimental and statistical methods. The interpretation of these results is not straightforward, however, such that they currently only provide **incomplete** support for the claims.

## Introduction

Interoceptive sensory input ascending from visceral organs – such as the heart, lungs, and gut – impacts both spontaneous brain activity and that evoked by external stimuli (*Azzalini et al., 2019*; *Critchley and Harrison, 2013*). Cardiac effects on exteroceptive perception have recently become a subject of intense interest, due in no small part to recent theories that interoception plays an integral role in generating the subjective experience of self (*Park and Tallon-Baudry, 2014*; *Seth and Tsakiris, 2018*). Early work in this area resulted in the Baroreceptor Hypothesis (*Lacey and Lacey, 1978*), which states that baroreceptors located in the arterial walls respond to the increase in blood pressure following cardiac contraction, and the ascending input from these baroreceptors broadly suppress sensorimotor activity in the central nervous system. While recent reviews have noted the Baroreceptor Hypothesis still motivates much work in the study of interoceptive–exteroceptive integration (*Park and Tallon-Baudry, 2014*), more recent studies have indicated, in

 

contrast, that visual skills such as discrimination (*Pramme et al., 2014*) and search (*Pramme et al., 2016*) are *facilitated* when stimuli are presented during cardiac systole. Work focused specifically on basic visual awareness rather than visual skills, however, has indeed continued to support the view that conscious access to a visual stimulus is suppressed during systole, and that this suppression is mediated by the insula, which is the primary thalamocortical recipient of baroreceptive input (*Salomon et al., 2016*; *Salomon et al., 2018*). This discrepancy may be taken to suggest that cardiac inputs to the central nervous system (CNS) are initially suppressive for the earliest stages of visual processing, consistent with the Baroreceptor Hypothesis, but have more heterogenous effects downstream, enabling more complex 'cardiac gating' effects such as systolic facilitation of threat (*Garfinkel et al., 2014*) and disgust responses (*Gray et al., 2012*) to visual stimuli. If so, however, the level of processing at which cardiac effects on visual perception may shift from suppressive to faciliatory remains unclear.

Resolving this ambiguity seems essential for understanding the proposed link between intero-ceptive–exteroceptive integration and self-awareness (*Park and Tallon–Baudry, 2014*; *Seth and Tsakiris, 2018*). Synchronizing visual stimuli to the heartbeat seems to affect the subjective experience of body ownership, indicating an important role of bodily sensations in carving out which parts of the exteroceptive sensory space belong to oneself – one in which interoceptive sensations provide a 'ground truth' point of reference that must originate from one's own body. For example, causing the color of a virtual body to pulse during cardiac systole facilitates the subjective embodiment of a virtual hand (*Suzuki et al., 2013*), a full virtual body (*Aspell et al., 2013*; *Heydrich et al., 2018*), and of a face image (*Sel et al., 2017*). In turn, changes in perceived embodiment are reflected in the magnitude of the central neural response to heartbeats (*Park et al., 2016*). Similar effects on embodiment can be seen for breathing (*Monti et al., 2020*) and gastric sensations (*Monti et al., 2022*). Since, simplistically, the two main ingredients of self-awareness would seem to be *self* and *awareness*, uniform suppression of sensations synchronized to interoceptive stimuli – those that are embodied as *self* – would arguably hinder an organism's ability to become self-aware. However, whatever adaptive benefits of self-awareness may exist, they must be balanced against the imperative to suppress distracting sensory information from awareness, just as the cortical response to one's heartbeat sounds appears to be suppressed (*van Elk et al., 2014*). Indeed, bodily sensations generally tend to fade from awareness as one goes about their day unless attention is drawn to them, which itself seems to be a core phenomenological characteristic of bodily self-awareness (*Allen and Tsakiris, 2018*). However, it is worth noting that volitional conscious access to one's heartbeat sensations, while rare in the general population, is associated with improved emotional regulation and mental well-being, indicating that the ability to hold the direct sensory consequences of bodily events into consciousness can serve an adaptive function (*Füstös et al., 2013*; *Tsakiris and Critchley, 2016*). In light of these considerations, we posit that the earliest level of sensory processing at which cardiac systole may become, in some contexts, faciliatory should be close enough to the level at which suppression has previously been observed to counter suppressive effects as needed.

Recent work demonstrating suppressive effects of cardiac input on visual awareness has used continuous flash suppression (CFS), a special case of binocular rivalry (*Salomon et al., 2016*; *Salomon et al., 2018*), and visual crowding (*Salomon et al., 2016*) paradigms to mask stimuli from consciousness, and showed that the threshold for breaking through into awareness is higher in both cases when masked stimuli are entrained to cardiac systole. A commonality between both paradigms is that, at the beginning of each trial, the masked stimulus is just that – masked – and the trial ends when it breaks into awareness. However, theories of consciousness, particularly but not limited to the popular global neuronal workspace theory (*Mashour et al., 2020*), tend to emphasize not just a stimulus' ability to initially break into consciousness but its ability to suppress competing stimuli so as to remain in consciousness. This sort of visual competition is exemplified by the classic binocular rivalry paradigm, in which incongruent stimuli are presented to subjects' two different eyes, and the brain resolves the conflict by switching back-and-forth between the two competing images rather than merging them (*Carmel et al., 2010*). While reflecting the same mechanism of interocular suppression as the previously used CFS paradigms, the durations of dominance and suppression for the rival stimuli are the cumulative product of the feedforward drive ascending from the optic nerve, lateral inhibition between eyes, and feedback from higher-level visual or cross-modal areas such as in

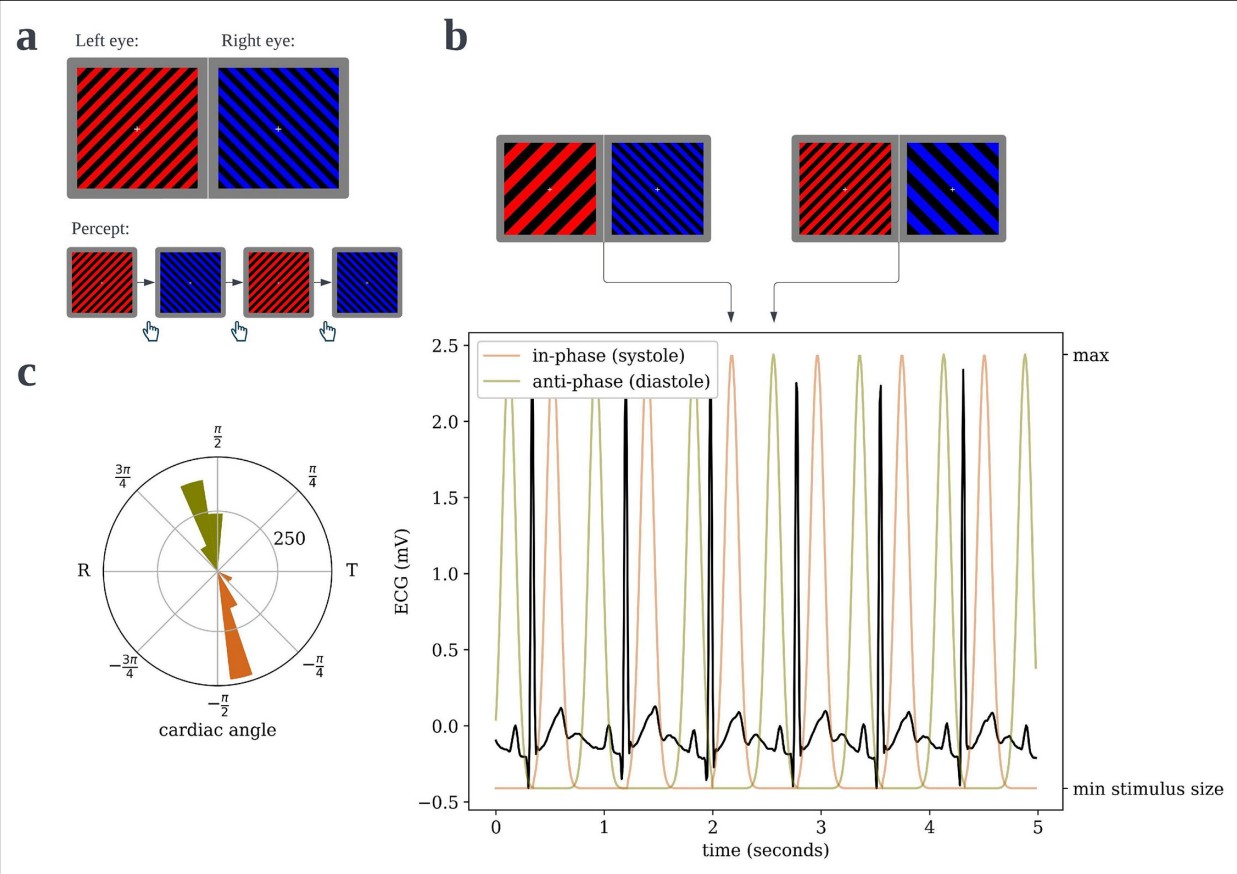

**Figure 1.** Cardiac binocular rivalry task. (**a**) As in a typical binocular rivalry paradigm, incongruent gratings were presented to the left and right eyes, causing the subjects' percept to alternate between the competing images. Each time the perceptually dominant image switched, subjects indicated so with a button press, and the durations of dominance were recorded. (**b**) The width of the bars in the grating would 'pulse', one in-phase and one anti-phase to the period of maximal blood pressure and thus maximal aortic baroreceptor activity following ejection of blood from the heart. (**c**) An example circular histogram of the cardiac phases/angles at which grating pulses peak over a full 10-min block, as recorded from an actual subject. Angles are scaled such that the R-peak is always at π radians and the offset of the T-wave is at 0 radians, thus the lower half of the circle is cardiac systole and the upper half is diastole.

directed attention (*Zhang et al., 2011*). In contrast, CFS breakthrough times are theoretically influenced primarily by modulation of the feedforward drive, rather than by lateral inhibition or consciously directed attention.

Thus, we designed an experiment in which we entrained dynamically 'pulsing' rivalrous stimuli to systolic and diastolic cardiac phases, cueing pulses based on subjects' electrocardiogram (ECG) in real time. The Baroreceptor Hypothesis would predict that the stimulus entrained to systole would spend more time suppressed and, conversely, less time dominant, as cortical activity would be suppressed each time that stimulus pulses. Indeed, such a finding would be consistent with the previously demonstrated (*Salomon et al., 2016*) and replicated (*Salomon et al., 2018*) suppression of stimuli entrained to cardiac systole in the CFS paradigm; this would suggest systolic modulatory effects do not become facilitative until a higher level of processing. An effect in the opposite direction – as we actually observe – would indicate that, at this early stage of visual processing, lateral inhibitory or top–down feedback mechanisms are sufficient to overcome the suppression effect observed in CFS. To differentiate between these two mechanisms, we then assess whether our main finding persists in those subjects who are at-chance at a heartbeat discrimination task (*Figure 1*).

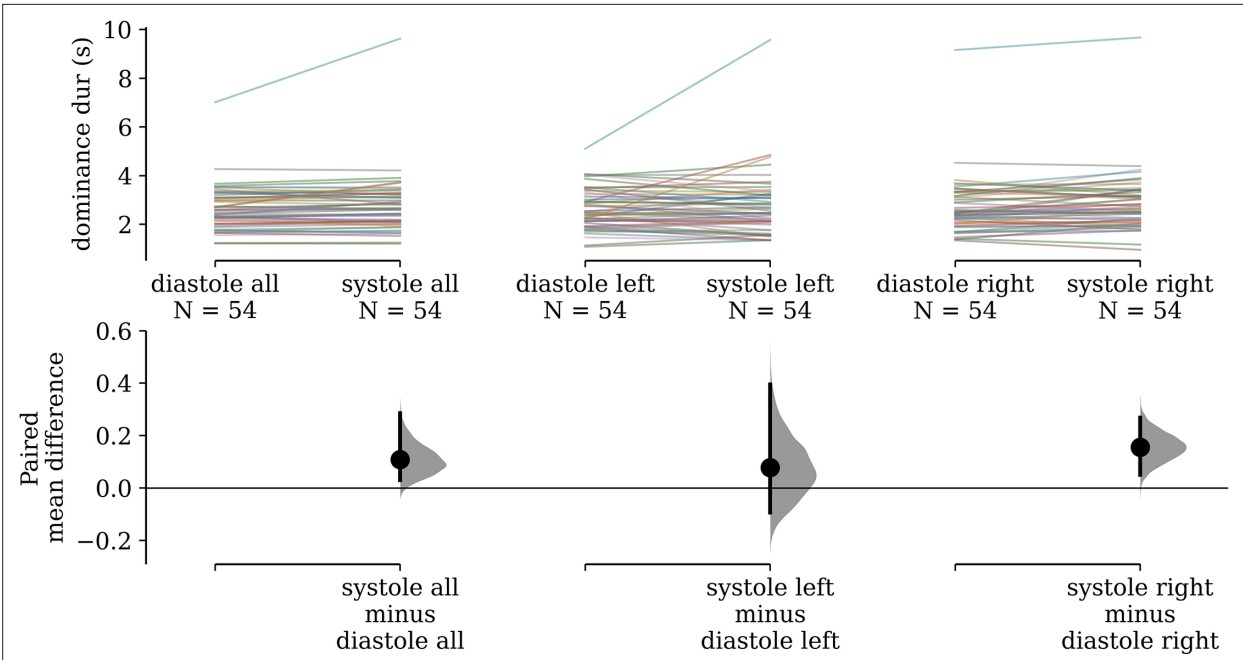

**Figure 2.** Paired differences in subjects' mean dominance durations. (Left) Subjects' mean dominance durations for grating stimuli that pulse in-phase with cardiac systole compared to those that pulse in diastole are shown on top. The colored lines denote individual subjects and connect their condition means. Bootstrap distributions and 95% confidence intervals of the paired differences between conditions are shown below. (Middle) The same visualization is shown for only stimuli presented to the left eye (Right) and only to the right eye.

## Results

### Entrainment to cardiac systole facilitates visual dominance

In the rivalry task, subject-wise mean dominance durations were 0.108 s (95% confidence interval [CI]: [0.030, 0.29]) longer for stimuli pulsing in-phase with cardiac systole than those anti-phase to systole (i.e. in diastole). These paired differences are visualized in *Figure 2*, and are also shown separately for stimuli presented to the left (*M* = 0.078, 95% CI: [−0.093, 0.395]) and the right (*M* = 0.156, 95% CI: [0.050, 0.268]) eyes, for readers who wish to assess for effects of baseline ocular dominance.

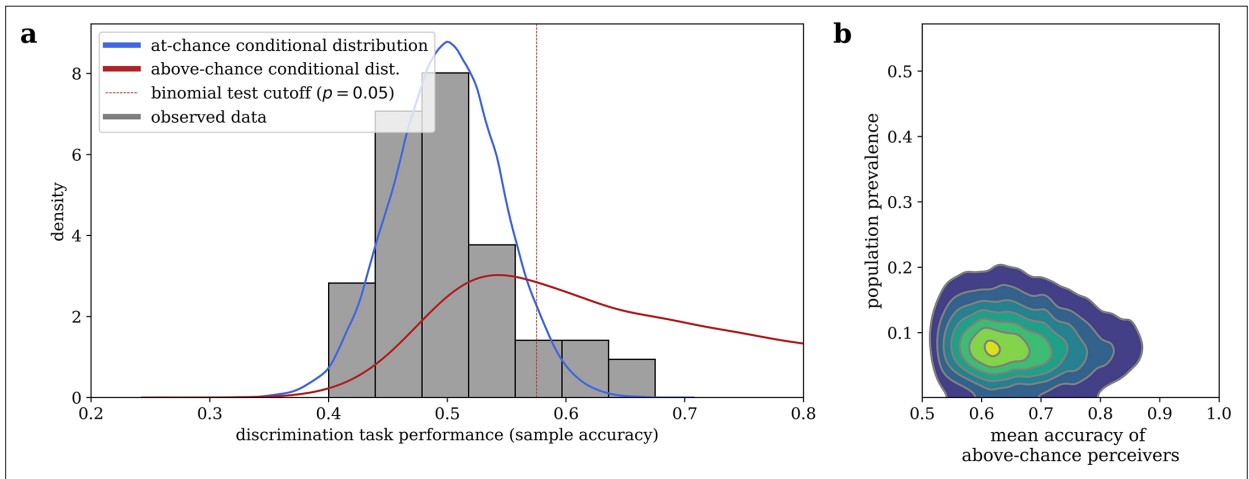

**Figure 3.** Bayesian mixture model of interoceptive accuracy. (**a**) Observed accuracies on the heartbeat discrimination task are modeled as a mixture between an at-chance Binomial distribution and an above-chance Beta-Binomial distribution, the conditional posterior predictive likelihoods of which are shown. (**b**) Conditional likelihoods can be combined with the estimated population prevalence of above-chance performers to assign of posterior probability of belonging to each distribution to any given subject.

Our estimated gamma generalized linear model (GLM) had a positive coefficient for the effect of synchronizing to cardiac systole (beta = 0.026, 95% CI: [0.002, 0.050]), which was statistically significant (Z = 2.157, p = 0.031). The value of this coefficient can be interpreted to mean that we estimate a 2.64% increase (95% CI: [0.24, 5.09]) in dominance duration for systole-entrained over diastole-entrained stimuli, on average.

## Cardiac facilitation of dominance does not depend on awareness of heartbeat

Our mixture model, visualized in *Figure 3*, estimated that 9.48% (95% highest desnity interval i.e. HDI: [1.81, 18.4]) of the sampled population are above-chance cardiac perceivers, and that above-chance perceivers have a mean accuracy of 0.668 (95% HDI: [0.521, 0.847]). Note that, while the estimated 9.48% population prevalence closely agrees with the 9.26% of subjects who were significantly above-chance via a Binomial test in our sample, the mixture model still assigns substantial probability to an above-chance perceiver performing worse than that frequentist cutoff (see *Figure 3*), illustrating why interpreting a frequentist null result would be inappropriate in this case.

When we remove subjects with greater than a 0.05 probability of being an above-chance perceiver from our sample, 46 subjects (85.2%) remain. This amounted to an effective cutoff of 0.542 accuracy. When we repeat the GLM analysis from the above section on just these subjects that we are quite sure are at-chance, we still find an effect of entrainment to cardiac systole on dominance durations (beta = 0.028, 95% CI: [0.001, 0.055], Z = 2.021, p = 0.043). The fact that the rivalry result can be found in a group of only cardiac non-perceivers indicates that the effect does not depend on subjects' conscious access to their heartbeat sensations, and thus also not on some conscious inference subjects may have made about which stimulus they are meant to attend to.

## Manipulation check

For all subjects, bootstrap mean cardiac angles of the stimulus 'pulse' times were negative (i.e. peaking in systole) for the in-phase stimulus and positive (i.e. in diastole) for the anti-phase stimulus, indicating our manipulation of cardiac phase was successful. The group mean cardiac angle for the systole-entrained pulses was −1.04 (95% CI: [−1.10, −0.99]) and for anti-phase pulses was 1.93 (95% CI: [1.88, 1.98]).

## Discussion

In light of previous work demonstrating that visual stimuli presented during cardiac systole are less likely to break through interocular suppression initially (*Salomon et al., 2016*; *Salomon et al., 2018*), presumably reflecting suppression of the feedforward drive ascending from the optic nerve through low-level visual areas, we suggest the faciliatory effect of systolic entrainment on interocular dominance observed in our rivalry paradigm is likely accounted for primarily by lateral inhibition between neurons responsive to each eye. In such an interpretation, this lateral inhibition is what would potentiated by baroreceptive input in our rivalry task. Interestingly, this could occur merely as a function of lateral inhibition occurring simultaneously to the broad cortical suppression conjectured in the original version of the Baroreceptor Hypothesis, increasing the total amount of inhibitory neurotransmitter in the post-synaptic cleft (*Lacey and Lacey, 1978*). Thus, the same broad cortical suppression could have a suppressive or faciliatory effect on any given stimulus *depending on context*.

However, this explanation – while intriguing – raises the very issue that prevents us from making such an inference by comparison to past cardio-visual studies that used CFS. In the CFS paradigm used in previous work (*Salomon et al., 2016*; *Salomon et al., 2018*), the mask stimulus is updated at a rate of 10 Hz to maintain suppression over time, and so new mask stimuli do appear several times each systolic phase. If lateral inhibition is indeed potentiated during systole as we suggest, and the mask stimulus is always dominant at trial start (which it is), then previous CFS findings could be re-interpreted as reflecting facilitation of the mask stimulus via lateral inhibition rather than direct suppression of the systole-entrained target stimulus via insula-to-visual projections. While this possibility seems to be made less likely by the fact that similar suppression of the systole-entrained stimulus is also observed in visual crowding (*Salomon et al., 2016*), the point remains the same as above: the same neural mechanism *could* result in either facilitation or suppression of a given sensory stimulus,

depending on context. Indeed, such context dependency could potentially explain the seemingly conflicting reports from previous studies on cardio-visual interaction (*Critchley and Harrison, 2013*). It may be fruitful for future work to compare cardiac effects across CFS and traditional rivalry paradigms with accompanying neural recordings.

Interestingly, the faciliatory effect we see on visual dominance does not appear to depend on conscious access to one's heartbeat sensations, as removing subjects who are above-chance at discriminating which stimulus is synchronized to their heartbeat from our sample does not eliminate the effect. Many effects of cardiac timing on perception and cognition *are* modulated by conscious interoceptive accuracy or awareness (*Koreki et al., 2022*; *Sel et al., 2017*), so it would be interesting to test whether interoceptive awareness potentiates the effect further. However, since we only had a handful of subjects in our sample that we could confidently conclude were above-chance heartbeat perceivers, this study cannot answer such a question with any statistical certainty.

Lastly, this finding may have implications for how we interpret the relationship between visual awareness and bodily self-awareness. A growing body of evidence suggests that visual self-recognition can occur in the absence of conscious awareness, as self-selective neural and behavioral responses can be observed for stimuli presented subliminally (*Bola et al., 2021*; *Doradzińska et al., 2020*; *Geng et al., 2012*; *Ota and Nakano, 2021*; *Pfister et al., 2012*; *Shafto and Pitts, 2015*) or too early after a stimulus event for subjects to have yet become consciously aware (*Veillette et al., 2023*). If so, self-awareness may result from a first-order awareness of self-referential cognitive processes (*some* of which need not depend on conscious awareness to function), as previously suggested as a possibility by some researchers (*Dehaene, 2014*), rather than a higher-level sort of 'awareness of being aware' as it may be understood by some cognitive scientists or in popular culture. In such a view – which, to be clear, may be incorrect – then any common factor which facilitates both awareness and perceived embodiment, even if by separate pathways, could be a contributing mechanism that facilitates self-awareness. Considering the present results in light of previous work on embodiment (*Aspell et al., 2013*; *Heydrich et al., 2018*; *Sel et al., 2017*), interoceptive–exteroceptive synchrony seems as if it might meet both those criteria.

## Methods

### Recruitment

We recruited 58 subjects (see *Sample size determination*) with self-reported normal color vision from our Psychology Department's subject pool, which consists of undergraduate students. Two subjects were dropped due to a technical error during data collection, and two subjects were dropped for failing to comply with task instructions, resulting in a final sample size of $n = 54$ subjects (15 biological males, 39 females, ages 20.4 ± 1.1 standard deviation).

### ECG acquisition and real-time processing

ECG was acquired at a 100-Hz sampling rate using a TMSi SAGA amplifier (TMSi, Netherlands) and a bipolar montage, with one electrode placed under the right clavicle, one at the bottom of the left ribcage, and ground under the left clavicle. Real-time analysis of the ECG data was implemented in Python (see *Data availability*) using Lab Streaming Layer (LSL, https://labstreaminglayer.org/#/) and LabGraph (https://github.com/facebookresearch/labgraph, *Facebook Reality Labs, 2021*). After digitizing, ECG data were bandpass filtered to 5–15 Hz, and R-peaks were detected using the Pan-Thompkins algorithm (*Pan and Tompkins, 1985*), modified from an existing implementation for LabGraph compatibility (*Sznajder and Łukowska, 2017*).

From the estimated R-peak times, we generated stimulus time courses on-the-fly as follows (in both the binocular rivalry task and the cardiac discrimination task). Since our ECG amplifier has a known analog-to-digital conversion latency of roughly 35 ms when acquiring data over LSL, we aimed for pulses of the in-phase/systole stimulus to peak at 175 ms, such that it would actually peak at 210 after each R-peak, following previous work on cardiac synchrony effects on embodiment (*Suzuki et al., 2013*). This timing coincides with peak systolic blood pressure, when the aortic baroreceptors are most active, and when heartbeats tend to be perceived in interoceptive sensitivity tasks (*Brener et al., 1993*; *Wiens and Palmer, 2001*). The anti-phase/diastole stimulus was delayed relative to the in-phase stimulus by half of the previous R–R interval. Since this delay was variable between heartbeats – thus

distinct from a constant phase delay but, on average, still anti-phase. This avoided the impression that one stimulus was leading or lagging, but as a result, the precise cardiac phase (i.e. within diastole) that the anti-phase stimulus peaked varied across subjects, but it nonetheless peaked during diastole for all subjects (see *ECG offline processing and analysis*). Stimulus 'pulses' had the shape of a Gaussian bell curve with a scale (i.e. 'standard deviation') of 1/16 of a second (see *Figure 1*). The target stimulus size was updated each time a new ECG sample was acquired (100 Hz), and total processing time to compute the intended stimulus time series was roughly a millisecond. On each screen refresh (60 Hz), the actual stimuli were updated to match their current target size.

## Cardiac binocular rivalry task

Subjects wore red-blue anaglyph glasses, such that a red image could be presented exclusively to the left eye and a blue image could be presented only to the right eye. The task consisted of two 10-min blocks, in which incongruent gratings (256 × 256 pixels on a 1920 × 1080, 22' monitor, which subjects viewed from 2 feet away) were presented to opposite eyes, which is well known to result in the centrally perceived image alternating between the two gratings – rather than, say, merging them (*Carmel et al., 2010*). As in a typical binocular rivalry paradigm, the subjects indicated with a button press each time the dominant stimulus switched. One grating would 'pulse' – the bars in the grating would briefly increase in width before returning to baseline, see *Figure 1* – during cardiac systole and one would pulse anti-phase to the systole pulse, during cardiac diastole (see *ECG acquisition and real-time processing*). In one 10-min block, the red/left grating would be the systole/in-phase stimulus and the blue/right grating would be the diastole/anti-phase stimulus, and in the other block it would be reversed; block order was randomized. The durations during which each grating was dominant were recorded for analysis (see *Behavioral data analysis*). Subjects were not told that the movement of the stimuli was related to their heartbeat until after this block had ended, and the heartbeat discrimination task had begun.

## Heartbeat discrimination task

Our modified heartbeat discrimination task was designed to assess whether subjects would be able to discriminate which of the two stimuli in the rivalry task was entrained to systole versus to diastole, rather than to assess more general interoceptive ability. Thus, we ignored some of the recent developments in improving the validity of heartbeat discrimination tasks for assessing the latter (*Brener and Ring, 2016*), and we used a bespoke task in which the manipulation closely matched that in our rivalry task. In particular, subjects saw two circles side-by-side on their screen, and the circles 'pulsed' (i.e. transiently increased in radius from 60 to 120 pixels) following the same time course as the gratings in the rivalry task – that is, one during systole and one during diastole (see *ECG acquisition and real-time processing* and *Figure 1*). On each of 120 10-s trials, the side of the systole stimulus and that of the diastole stimulus was randomized, and subjects were asked to report which circle was pulsing in synchrony with their heart. The accuracy of each response was recorded for analysis (see *Behavioral data analysis*).

## Behavioral data analysis

For visualization and estimation of paired differences, the dominance durations for the systole- and diastole-entrained gratings were averaged across all trials and block within each subject. Those measurements, and the bootstrap CIs (10,000 samples) of their paired differences were plotted using the *DABEST* Python package for robust estimation statistics (*Ho et al., 2019*). We additionally visualize these measurements broken up by whether the grating was presented to the left or the right eye, in case of baseline ocular dominance confounds.

For statistical inference on dominance durations, we apply a gamma-family GLM with a dummy predictor for cardiac systole to the trial-level data (i.e. each interval, unaveraged), fit using generalized estimating equations in the *statsmodels* package to account for subject-level random effects (*Rotnitzky and Jewell, 1990*; *Seabold and Perktold, 2010*). This approach appropriately deals with the approximate gamma distribution that dominance durations in binocular rivalry are known to follow, while remaining robust to violations of parametric assumptions (*Carmel et al., 2010*).

We then re-ran the above GLM analysis on only those subjects who were at-chance at the heartbeat discrimination task, so we could assess whether the result from the first GLM would hold for only

subjects who could not consciously distinguish which stimulus was in-phase with cardiac systole – and thus whether our results depend on conscious awareness of heartbeat sensations. Since identification of 'at-chance' subjects would require interpretation of a null result in a frequentist setting (so we cannot just use a Binomial test), we estimate the probability each subject is at- or above-chance using a Bayesian mixture model, and then we call subjects 'at-chance' if there is at least a 95% posterior probability they are indeed drawn from the at-chance distribution (see below).

In the mixture model, each subject $i$'s number of correct trials $k_i$ is modeled as coming from either an at-chance distribution $k_i|\text{at-chance} \sim \text{Binomial}\,(n_{\text{trials}}, 0.5)$ or an above-chance distribution $k_i|\text{above-chance} \sim \text{Binomial}\,(n_{\text{trials}}, p_i)$ where $p_i \sim 0.5 + \text{Beta}\,(\alpha, \beta)\,/2$, which spans the range $[0.5, 1.0]$. We pick priors for parameters $\alpha \sim \text{Exponential}\,(1.0)$ and $\beta \sim \text{Exponential}\,(0.5)$, which slightly favors distributions where accuracy $p_i$ is more likely to be close to 0.5 than to 1.0, and we put a $\text{Uniform}\,(0, 1)$ prior on the population prevalence of above-chance perceivers. Given posteriors for these parameters, each subject's probability of coming from one distribution or another given their observed performance $k_i$ is simply given by Bayes' rule. Posterior distributions were approximated with 10,000 posterior samples in *PyMC* (*Patil et al., 2010*).

### ECG offline processing and analysis

While characteristics of the ECG data were not of particular interest in the present study outside of our online manipulation of stimulus timing (see *ECG acquisition and real-time analysis*), we did estimate the cardiac phase at which entrained stimuli actually peaked in the rivalry task as an offline manipulation check. (For instance, if the anti-phase stimulus did not peak during diastole, which could occur for a subject with a very fast heart rate, it might be prudent to exclude that subject from analysis – though this did not end up occurring, see *Results*.)

To this end, we applied the standardized NeuroKit pipeline, as implemented in version 0.2.7 of the *neurokit2* Python package, to delineate cardiac events in the ECG data from both blocks of the rivalry task (*Makowski et al., 2021*). In brief, this applies a 0.5-Hz high pass filter to the data, detects QRS complexes (i.e. the Q, R, and S waves that represent the deporalization of the ventricles and contraction of the ventricular muscles) based on the steepness of the absolute derivative of the signal, then R-peaks (i.e. the start of systole) as the local maxima in the QRS complexes, and finally the offset of the T-wave (i.e. end of ventricular systole and the start of diastole) using the discrete wavelet method (*Martínez et al., 2004*). We then compute the 'cardiac angle' at each peak of the entrained stimuli using the same physiologically motivated scaling implemented in the Cardiac Timing Toolbox (*Sherman et al., 2022*); phases of the cardiac cycle are expressed as an angle where $\pi$ radians is always the R-peak and 0 radians is the offset of the T-wave, such that 'negative' angles in the range $[-\pi, 0]$ occur during systole and 'positive' angles in the range $[0, \pi]$ occur during diastole.

We then obtain (circular) means for the cardiac angle the synchronous and asynchronous stimuli peak for each subject by bootstrap with 10,000 samples. These are used for a manipulation check to confirm that the mean angle for the systole stimulus is negative for all subjects, and the mean angle for the diastole stimulus is positive for all subjects. We then estimate the group-level mean cardiac angles for both stimuli using a random-effects bootstrap scheme, which accounts for both the population sampling distribution and the within-subject sampling distribution (*Chambers and Chandra, 2013*). Group-level bootstrap means and CIs are reported in results.

### Sample size determination

Our sample size was initially determined arbitrarily (by time constraints on data collection). However, we did conduct a power calculation to assess whether this convenience sample was sufficiently large to detect a typical cardio-visual effect. To this end, we used the smallest standardized effect size (paired Cohen's *d* = 0.38) reported across the seven cardio-visual experiments reported by *Salomon et al., 2016*. This calculation yielded a power of 0.81 at a significance level of 0.05 for our original *n* = 58 subjects and 0.78 for our final *n* = 54 subjects, which we deemed sufficient to stop data collection.

## Acknowledgements

This work was supported by NSF BCS 2024923 to HCN; JPV was supported by NSF DGE 1746045.

## Additional information

### Funding

| Funder | Grant reference number | Author |
|---|---|---|
| National Science Foundation | 2024923 | Howard C Nusbaum |
| National Science Foundation | Graduate Research Fellowship Program (1746045) | John P Veillette |

The funders had no role in study design, data collection, and interpretation, or the decision to submit the work for publication.

### Author contributions

John P Veillette, Conceptualization, Data curation, Software, Formal analysis, Funding acquisition, Validation, Investigation, Visualization, Methodology, Writing - original draft, Project administration, Writing - review and editing; Fan Gao, Data curation, Investigation, Writing - review and editing; Howard C Nusbaum, Resources, Supervision, Funding acquisition, Writing - review and editing

### Author ORCIDs

John P Veillette ⬛ https://orcid.org/0000-0002-0332-4372

### Ethics

Informed consent was obtained from all participants. Study procedures were approved by the Social and Behavioral Sciences Institutional Review Board (IRB23-1353) at the University of Chicago.

Reviewer #1 (Public Review): https://doi.org/10.7554/eLife.95599.2.sa1
Reviewer #2 (Public Review): https://doi.org/10.7554/eLife.95599.2.sa2
Reviewer #3 (Public Review): https://doi.org/10.7554/eLife.95599.2.sa3
Author response https://doi.org/10.7554/eLife.95599.2.sa4

## Additional files

### Supplementary files
• MDAR checklist

### Data availability

The raw data, organized according to the standardized Brain Imaging Data Structure (BIDS), is available on the Open Science Framework (https://doi.org/10.17605/OSF.IO/6ZMU8). The complete code for the experiment (https://doi.org/10.5281/zenodo.10367327) and data analysis (https://zenodo.org/doi/10.5281/zenodo.10367248) are permanently archived on Zenodo.

The following datasets were generated:

| Author(s) | Year | Dataset title | Dataset URL | Database and Identifier |
|---|---|---|---|---|
| Veillette J, Gao F, Nusbaum H | 2023 | ecg-rivalry | https://doi.org/10.17605/OSF.IO/6ZMU8 | Open Science Framework, 10.17605/OSF.IO/6ZMU8 |
| Veillette J | 2023 | apex-lab/ecg-rivalry-analysis: v0.0.1: for preprint | https://zenodo.org/records/10367249 | Zenodo, 10.5281/zenodo.10367248 |
| Veillette J | 2023 | apex-lab/ecg-rivalry-experiment: v1.0.0: Used to run experiment | https://zenodo.org/records/10367327 | Zenodo, 10.5281/zenodo.10367326 |

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
