## [Editor Report · eLife assessment]

This is a binocular rivalry study that uses ECG to present visual stimuli pulsing in line with cardiac events, to examine whether systole-entrained stimuli (i.e. presented during the period where the heart has contracted) are suppressed within visual awareness. Arguably out of line with this idea, the dominance durations were increased for systole-entrained stimuli. The manuscript addresses an **important**, precisely defined, and theoretically well-motivated question using sophisticated experimental and statistical methods. The interpretation of these results is not straightforward, however, such that they currently only provide **incomplete** support for the claims.

---

## [Referee Report · Reviewer #1 (Public Review)]

Summary:

The aim of the study described in this paper was to test whether visual stimuli that pulse synchronously with the systole phase of the cardiac cycle are suppressed compared with stimuli that pulse in the diastole phase. To this end, the authors employed a binocular rivalry task and used the duration of the perceived image as the metric of interest. The authors predicted that if there was global suppression of the visual stimulus during systole then the durations of the stimulus that were pulsing synchronously with systole should be of shorter duration than those pulsing in diastole. However, the results observed were the opposite of those predicted. The authors speculate on what this facilitation effect might mean for the baroreceptor suppression hypothesis.

Strengths:

This is an interesting and timely study that uses a clever paradigm to test the baroreceptor suppression hypothesis in vision. This is a refreshingly focussed paper with interesting and seemingly counterintuitive results.

Weaknesses:

The paper could benefit from a clearer explanation of the predicted results. For those not experts in binocular rivalry, it would be useful to explain the predicted results. Does pulsing stimuli in this way change durations in such a task? If there is global suppression of visual stimuli why would this lead to shorter/longer durations in the systole compared to the diastole conditions? In addition, the duration lengths in both conditions seem to be longer than one cardiac cycle. If the cardiac cycle modulates duration it would be interesting to discuss why this occurs on some cycles but not on others. If there is a facilitation effect why does it only occur on some cycles?

---

## [Referee Report · Reviewer #2 (Public Review)]

Summary:

This is a binocular rivalry study that uses electrocardiogram events to modulate visual stimuli in real-time, relative to participants' heartbeats. The main finding is that modulations during the period around when the heart has contracted (systole) increase rivalry dominance durations. This is a really neat result, that demonstrates the link between interoception and vision. I thought the Bayesian mixture modelling was a really smart way to identify cardiac non-perceivers, and the finding that the main result is preserved in this group is compelling. Overall, the study has been conducted to a high standard, is appropriately powered, and reported clearly. I have one suggestion about interpretation, which concerns the explanation of increased dominance durations with reference to contemporary models of binocular rivalry, and a few minor queries. However, I think this paper is a worthwhile addition to the literature.

---

## [Referee Report · Reviewer #3 (Public Review)]

Summary:

The manuscript addresses a question inspired by the Baroceptor Hypothesis and its links to visual awareness and interoception. Specifically, the reported study aimed to determine if the effects of cardiac contraction (systole) on binocular rivalry (BR) are facilitatory or suppressive. The main experiment - relying on a technically challenging procedure of presenting stimuli synchronised with the heartbeats of participants - has been conducted with great care, and numerous manipulation checks the authors report convincingly show that the methods they used work as intended. Moreover, the control experiment allows for excluding alternative explanations related to participants being aware of their heartbeats. Therefore, the study convincingly shows the effect of cardiac activity on BR - and this is an important finding. The results, however, do not allow for unambiguously determining if this effect is facilitatory or suppressive (see details below), which renders the study not as informative as it could be.

While the authors strongly focus on interoception and awareness, this study will be of interest to researchers studying BR as such. Moreover, the code and the data the authors share can facilitate the adoption of their methods in other labs.

Strengths:

(1) The study required a complex technical setup and the manuscript both describes it well and demonstrates that it was free from potential technical issues (e.g. in section 3.3. Manipulation check).

(2) The sophisticated statistical methods the authors used, at least for a non-statistician like me, appear to be well-suited for their purpose. For example, they take into account the characteristics of BR (gamma distributions of dominance durations). Moreover, the authors demonstrate that at least in one case their approach is more conservative than a more basic one (Binomial test) would be.

(3) Finally, the control experiment, and the analysis it enabled, allow for excluding a multitude of alternative explanations of the main results.

(4) The authors share all their data and materials, even the code for the experiment.

(5) The manuscript is well-written. In particular, it introduces the problem and methods in a way that should be easy to understand for readers coming from different research fields.

Weaknesses:

(1) The interpretation of the main result in the context of the Baroceptor hypothesis is not clear. The manuscript states: The Baroreceptor Hypothesis would predict that the stimulus entrained to systole would spend more time suppressed and, conversely, less time dominant, as cortical activity would be suppressed each time that stimulus pulses. The manuscript does not specify why this should be the case, and the term 'entrained' is not too helpful here (does it refer to neural entrainment? or to 'being in phase with'?). The answer to this question is provided by the manuscript only implicitly, and, to explain my concern, I try to spell it out here in a slightly simplified form.

During systole (cardiac contraction), the visual system is less sensitive to external information, so it 'ignores' periods when the systole-synchronised stimulus is at the peak of its pulse. Conversely, the system is more sensitive during diastole, so the stimulus that is at the peak of its pulse then should dominate for longer, because its peaks are synchronised with the periods of the highest sensitivity of the visual system when the information used to resolve the rivalry is sampled from the environment. This idea, while indeed being a clever test of the hypothesis in question, rests on one critical assumption: that the peak of the stimulus pulse (as defined in the manuscript) is the time when the stimulus is the strongest for the visual system. The notion of 'stimulus strength' is widely used in the BR literature (see Brascamp et al., 2015 for a review). It refers to the stimulus property that, simply speaking, determines its tendency to dominate in the BR. The strength of a stimulus is underpinned by its low-level visual properties, such as contrast and spatial frequency content. Coming back to the manuscript, the pulsing of the stimuli affected at least spatial frequency (and likely other low-level properties), and it is unknown if it was in phase with the pulsing of the stimulus strength, or not. If my understanding of the premise of the study is correct, the conclusions drawn by the authors stand only if it was.

In other words, most likely the strength of one of the stimuli was pulsating in sync with the systole, but is it not clear which stimulus it was. It is possible that, for the visual system, the stimulus meant to pulse in sync with the systole was pulsing strength-wise in phase with the diastole (and the one intended to pulse with in sync with the diastole strength-wise pulsed with the systole). If this is the case, the predictions of the Baroceptor Hypothesis hold, which would change the conclusion of the manuscript.

(2) Using anaglyph goggles necessitates presenting stimuli of a different colour to each eye. The way in which different colours are presented can impact stimulus strength (e.g. consider that different anaglyph foils can attenuate the light they let through to different degrees). To deal with such effects, at least some studies on BR employed procedures of adjusting the colours for each participant individually (see Papathomas et al., 2004; Patel et al., 2015 and works cited there). While I think that counterbalancing applied in the study excludes the possibility that colour-related effects influenced the results, the effects of interest still could be stronger for one of the coloured foils.

(3) Several aspects of the methods (e.g. the stimuli), are not described at the level of detail some readers might be accustomed to. The most important issue here is the task the participants performed. The manuscript says that they pressed a button whenever they experienced a switch in perception, but it is only implied that there were different buttons for each stimulus.

Brascamp, J. W., Klink, P. C., & Levelt, W. J. M. (2015). The 'laws' of binocular rivalry: 50 years of Levelt's propositions. Vision Research, 109, 20-37. https://doi.org/10.1016/j.visres.2015.02.019

Papathomas, T. V., Kovács, I., & Conway, T. (2004). Interocular grouping in binocular rivalry: Basic attributes and combinations. In D. Alais & R. Blake (Eds.), Binocular Rivalry (pp. 155-168). MIT Press

Patel, V., Stuit, S., & Blake, R. (2015). Individual differences in the temporal dynamics of binocular rivalry and stimulus rivalry. Psychonomic Bulletin and Review, 22(2), 476-482. https://doi.org/10.3758/s13423-014-0695-1

---

## [Author Response]

We thank the reviewers for their productive comments on our work. While we have chosen to not revise the manuscript further, we reply to the public reviewer comments here so as to provide clarification on certain points.

**Reviewer #1 (Public Review):**
Summary:The aim of the study described in this paper was to test whether visual stimuli that pulse synchronously with the systole phase of the cardiac cycle are suppressed compared with stimuli that pulse in the diastole phase. To this end, the authors employed a binocular rivalry task and used the duration of the perceived image as the metric of interest. The authors predicted that if there was global suppression of the visual stimulus during systole then the durations of the stimulus that were pulsing synchronously with systole should be of shorter duration than those pulsing in diastole. However, the results observed were the opposite of those predicted. The authors speculate on what this facilitation effect might mean for the baroreceptor suppression hypothesis.Strengths:This is an interesting and timely study that uses a clever paradigm to test the baroreceptor suppression hypothesis in vision. This is a refreshingly focussed paper with interesting and seemingly counterintuitive results.Weaknesses:The paper could benefit from a clearer explanation of the predicted results. For those not experts in binocular rivalry, it would be useful to explain the predicted results. Does pulsing stimuli in this way change durations in such a task? If there is global suppression of visual stimuli why would this lead to shorter/longer durations in the systole compared to the diastole conditions? In addition, the duration lengths in both conditions seem to be longer than one cardiac cycle. If the cardiac cycle modulates duration it would be interesting to discuss why this occurs on some cycles but not on others. If there is a facilitation effect why does it only occur on some cycles?

In general, pulsing stimuli (i.e. moving gratings) show longer dominance durations when in competition with non-pulsing stimuli; in other words, pulses increase the “stimulus strength” of a visual grating (Wade, De Weert & Swanston, 1984). The Baroreceptor Hypothesis predicts global suppression of visual cortex during systole (and not during diastole), so the stimulus strength boost yielded by a pulse should be attenuated during systole. Thus, the stimulus that only pulses during systole would have lower stimulus strength (and thus shorter dominance durations) than that which pulses during diastole; however, we observe the opposite pattern in our data, seemingly contradicting the Baroreceptor Hypothesis.

In typical binocular rivalry paradigms, dominance durations are biased by stimulus strength, but perception remains bistable such that the stronger stimulus is not necessarily dominant at a given time. We see no reason, then, why switching would have to occur every cycle. The dominance durations we see are quite typical of binocular rivalry paradigms, whereas durations shorter than a cardiac cycle would be rather unusual (Carmel et al., 2010).

**Reviewer #2 (Public Review):**
Summary:This is a binocular rivalry study that uses electrocardiogram events to modulate visual stimuli in real-time, relative to participants' heartbeats. The main finding is that modulations during the period around when the heart has contracted (systole) increase rivalry dominance durations. This is a really neat result, that demonstrates the link between interoception and vision. I thought the Bayesian mixture modelling was a really smart way to identify cardiac non-perceivers, and the finding that the main result is preserved in this group is compelling. Overall, the study has been conducted to a high standard, is appropriately powered, and reported clearly. I have one suggestion about interpretation, which concerns the explanation of increased dominance durations with reference to contemporary models of binocular rivalry, and a few minor queries. However, I think this paper is a worthwhile addition to the literature.

The point Reviewer 2 makes with respect to contemporary models of binocular rivalry is important – perhaps more so than its brief statement in this public review suggests. As we already expand upon in our Discussion, the effects of global (neural) inhibition depend on the preexisting role that inhibition plays in a given neural circuit. The original framing of the Baroreceptor Hypothesis describes baroreceptor activity of uniformly impeding sensory processing (Lacey, 1967; Lacey & Lacey, 1978, American Psychologist), which is contradicted by our present results. This account is often interpreted as implying the effects of baroreceptor activation is inhibitory in terms of neural mechanism (e.g. Rau et al., 1993, Psychophysiology; Edwards et al., 2009, Psychophysiology). Some researchers argue this serves a parallel function to the inhibitory projections from motor to sensory areas during volitional movement, “cancelling” the sensory effects of heartbeats (Van Elk, et al., 2014, Biological Psychology).

However, baroreceptor activity has also been described as introducing noise into sensory processing rather than inhibiting it directly (e.g. Allen et al., 2022, PLoS Computational Biology). Lacey and Lacey’s own account actually seemed to point toward attention as a mediating mechanism (Hahn, 1973, Psychological Bulletin), with the disproportionate focus on cortical inhibition emerging in the literature over time. All this is to say that, while our results seem to falsify the behavioral predictions of the original Baroreceptor Hypothesis, subsequent versions of that hypothesis that describe an inhibitory neural mechanism, rather than an inhibition of perception per se, could potentially still be compatible with our results. This is a topic we plan to explore in future work.

**Reviewer #3 (Public Review):**
Summary:The manuscript addresses a question inspired by the Baroceptor Hypothesis and its links to visual awareness and interoception. Specifically, the reported study aimed to determine if the effects of cardiac contraction (systole) on binocular rivalry (BR) are facilitatory or suppressive. The main experiment - relying on a technically challenging procedure of presenting stimuli synchronised with the heartbeats of participants - has been conducted with great care, and numerous manipulation checks the authors report convincingly show that the methods they used work as intended. Moreover, the control experiment allows for excluding alternative explanations related to participants being aware of their heartbeats. Therefore, the study convincingly shows the effect of cardiac activity on BR - and this is an important finding. The results, however, do not allow for unambiguously determining if this effect is facilitatory or suppressive (see details below), which renders the study not as informative as it could be.While the authors strongly focus on interoception and awareness, this study will be of interest to researchers studying BR as such. Moreover, the code and the data the authors share can facilitate the adoption of their methods in other labs.Strengths:(1) The study required a complex technical setup and the manuscript both describes it well and demonstrates that it was free from potential technical issues (e.g. in section 3.3. Manipulation check).(2) The sophisticated statistical methods the authors used, at least for a non-statistician like me, appear to be well-suited for their purpose. For example, they take into account the characteristics of BR (gamma distributions of dominance durations). Moreover, the authors demonstrate that at least in one case their approach is more conservative than a more basic one (Binomial test) would be.(3) Finally, the control experiment, and the analysis it enabled, allow for excluding a multitude of alternative explanations of the main results.(4) The authors share all their data and materials, even the code for the experiment.(5) The manuscript is well-written. In particular, it introduces the problem and methods in a way that should be easy to understand for readers coming from different research fields.Weaknesses:(1) The interpretation of the main result in the context of the Baroceptor hypothesis is not clear. The manuscript states: The Baroreceptor Hypothesis would predict that the stimulus entrained to systole would spend more time suppressed and, conversely, less time dominant, as cortical activity would be suppressed each time that stimulus pulses. The manuscript does not specify why this should be the case, and the term 'entrained' is not too helpful here (does it refer to neural entrainment? or to 'being in phase with'?). The answer to this question is provided by the manuscript only implicitly, and, to explain my concern, I try to spell it out here in a slightly simplified form.During systole (cardiac contraction), the visual system is less sensitive to external information, so it 'ignores' periods when the systole-synchronised stimulus is at the peak of its pulse. Conversely, the system is more sensitive during diastole, so the stimulus that is at the peak of its pulse then should dominate for longer, because its peaks are synchronised with the periods of the highest sensitivity of the visual system when the information used to resolve the rivalry is sampled from the environment. This idea, while indeed being a clever test of the hypothesis in question, rests on one critical assumption: that the peak of the stimulus pulse (as defined in the manuscript) is the time when the stimulus is the strongest for the visual system. The notion of 'stimulus strength' is widely used in the BR literature (see Brascamp et al., 2015 for a review). It refers to the stimulus property that, simply speaking, determines its tendency to dominate in the BR. The strength of a stimulus is underpinned by its low-level visual properties, such as contrast and spatial frequency content. Coming back to the manuscript, the pulsing of the stimuli affected at least spatial frequency (and likely other low-level properties), and it is unknown if it was in phase with the pulsing of the stimulus strength, or not. If my understanding of the premise of the study is correct, the conclusions drawn by the authors stand only if it was.In other words, most likely the strength of one of the stimuli was pulsating in sync with the systole, but is it not clear which stimulus it was. It is possible that, for the visual system, the stimulus meant to pulse in sync with the systole was pulsing strength-wise in phase with the diastole (and the one intended to pulse with in sync with the diastole strength-wise pulsed with the systole). If this is the case, the predictions of the Baroceptor Hypothesis hold, which would change the conclusion of the manuscript.

We agree with Reviewer 3’s argumentation here. If the pulses decreased, rather than increased, effective stimulus strength, then the present results would indeed be consistent with the Baroreceptor Hypothesis. However, Wade et al. (1984) demonstrated that grating stimuli which pulse in the same manner (i.e. by dynamically varying the spatial frequency of the grating) as in our experiment indeed show increased stimulus strength relative to static stimuli, even if the dynamic stimuli have lower spatial frequency on average (https://doi.org/10.3758/BF03203891).

We admit our results would be stronger had we included a replication of Wade at al. (1984) in our study, but in light of this previous work, our interpretation is indeed supported.

(2) Using anaglyph goggles necessitates presenting stimuli of a different colour to each eye. The way in which different colours are presented can impact stimulus strength (e.g. consider that different anaglyph foils can attenuate the light they let through to different degrees). To deal with such effects, at least some studies on BR employed procedures of adjusting the colours for each participant individually (see Papathomas et al., 2004; Patel et al., 2015 and works cited there). While I think that counterbalancing applied in the study excludes the possibility that colour-related effects influenced the results, the effects of interest still could be stronger for one of the coloured foils.

It is the case that, when we split the data up by eye (and thus by color), we only see statistically significant results for one eye – though the nominal direction of the effect is consistent across both eyes. So it is indeed possible that the effect could be stronger for one of the colored foils, but the present experiment was not designed to be powered to test that cardiac phase-by-color interaction.

We concur with the Reviewer, however, that our use of counterbalancing excludes color-related effects as an explanation for our main findings.

(3) Several aspects of the methods (e.g. the stimuli), are not described at the level of detail some readers might be accustomed to. The most important issue here is the task the participants performed. The manuscript says that they pressed a button whenever they experienced a switch in perception, but it is only implied that there were different buttons for each stimulus.

There were indeed different buttons for each stimulus (i.e. a button to indicate their perception had switched to the red stimulus and another to indicate it had switched to blue). Our full, unmodified experiment code has been made available and is permanently archived (https://doi.org/10.5281/zenodo.10367327), so the full procedure is well documented and can be replicated exactly.

Brascamp, J. W., Klink, P. C., & Levelt, W. J. M. (2015). The 'laws' of binocular rivalry: 50 years of Levelt's propositions. Vision Research, 109, 20-37. https://doi.org/10.1016/j.visres.2015.02.019

Papathomas, T. V., Kovács, I., & Conway, T. (2004). Interocular grouping in binocular rivalry: Basic attributes and combinations. In D. Alais & R. Blake (Eds.), Binocular Rivalry (pp. 155-168). MIT Press

Patel, V., Stuit, S., & Blake, R. (2015). Individual differences in the temporal dynamics of binocular rivalry and stimulus rivalry. Psychonomic Bulletin and Review, 22(2), 476-482. https://doi.org/10.3758/s13423-014-0695-1